# Mir526b and Mir655 Promote Tumour Associated Angiogenesis and Lymphangiogenesis in Breast Cancer

**DOI:** 10.3390/cancers11070938

**Published:** 2019-07-04

**Authors:** Stephanie Hunter, Braydon Nault, Kingsley Chukwunonso Ugwuagbo, Sujit Maiti, Mousumi Majumder

**Affiliations:** Department of Biology, Brandon University, 3rd Floor, John R. Brodie Science Centre, 270–18th Street, Brandon, MB R7A 6A9, Canada

**Keywords:** miR526b, miR655, breast cancer, angiogenesis, lymphangiogenesis, *EP4*, PI3K/Akt

## Abstract

MicroRNAs (miRNAs) are small endogenously produced RNAs, which regulate growth and development, and oncogenic miRNA regulate tumor growth and metastasis. Tumour-associated angiogenesis and lymphangiogenesis are processes involving the release of growth factors from tumour cells into the microenvioronemnt to communicate with endothelial cells to induce vascular propagation. Here, we examined the roles of cyclo-oxygenase (COX)-2 induced miR526b and miR655 in tumour-associated angiogenesis and lymphangiogenesis. Ectopic overexpression of miR526b and miR655 in poorly metastatic estrogen receptor (ER) positive MCF7 breast cancer cells resulted in upregulation of angiogenesis and lymphangiogenesis markers vascular endothelial growth factor A (VEGFA); VEGFC; VEGFD; COX-2; lymphatic vessel endothelial hyaluronan receptor-1 (LYVE1); and receptors *VEGFR1*, *VEGFR2*, and *EP4*. Further, miRNA-high cell free conditioned media promoted migration and tube formation by human umbilical vein endothelial cells (HUVECs), and upregulated *VEGFR1,*
*VEGFR2,* and *EP4* expression, showing paracrine stimulation of miRNA in the tumor microenvironment. The miRNA-induced migration and tube formation phenotypes were abrogated with *EP4* antagonist or PI3K/Akt inhibitor treatments, confirming the involvement of the *EP4* and PI3K/Akt pathway. Tumour supressor gene *PTEN* was found to be downregulated in miRNA high cells, confirming that it is a target of both miRNAs. *PTEN* inhibits hypoxia-inducible factor-1 (HIF1α) and the PI3K/Akt pathway, and loss of regulation of these pathways through *PTEN* results in upregulation of VEGF expression. Moreover, in breast tumors, angiogenesis marker *VEGFA* and lymphangiogenesis marker *VEGFD* expression was found to be significantly higher compared with non-adjacent control, and expression of miR526b and miR655 was positively correlated with *VEGFA,*
*VEGFC,*
*VEGFD,*
*CD31,* and *LYVE1* expression in breast tumour samples. These findings further strengthen the role of miRNAs as breast cancer biomarkers and *EP4* as a potential therapeutic target to abrogate miRNA-induced angiogenesis and lymphangiogenesis in breast cancer.

## 1. Introduction

Breast cancer is the deadliest and most prevalent cancer among women, being responsible for the greatest number of cancer-related deaths among women worldwide [1]. In many cancers, including human breast cancer, cyclo-oxygenase (COX)-2 enzyme is found to have higher than normal expression [2]. Specifically, upregulation of COX-2 is correlated with breast cancer disease progression, metastasis, and poor patient survival [3,4]. COX-2 is responsible for the production of the inflammatory molecule, prostaglandin E2 (PGE2). Production of PGE2 by COX-2 results in binding of PGE2 to four G-protein coupled PGE receptors, EP1-4, each of which have distinct signalling pathways [5]. EP1 couples with Gq, EP3 couples with Gi, and EP2 and *EP4* couple with Gs. Additionally*, EP4* stimulates non-canonical pathways PI3K/Akt and ERK, which are associated with cell survival and migration [6]. PGE2 induces early vascular maturation and angiogenesis in vertebrates by upregulation of VEGFs and PGE2 receptors [7]. Overproduction of PGE2 and activation of *EP4* receptor results in many tumourigenesis-promoting phenotypes such as inactivation of host anti-tumour immune cells, enhanced tumour cell migration and invasion, stem-like cell (SLC) induction, and tumour-associated angiogenesis and lymphangiogenesis [6,8]. Overexpression of COX-2 in two poorly metastatic MCF7 (COX-2 low, human epidermal growth factor receptor 2 (HER-2) negative, progesterone receptor (PR) positive, estrogen receptor (ER) positive) and SK-BR-3 (COX-2 low, HER-2 high, ER negative) breast cancer cell lines has been shown to induce aggressive breast cancer phenotypes and promote metastasis, which could be abrogated with *EP4* antagonist treatment. Moreover, MCF7 cells show lower ER expression with COX-2 overexpression, and COX-2 overexpression caused a change in gene and microRNA (miRNA) expression in MCF7 cells [8].

MicroRNAs are a class of endogenously produced, short non-coding RNAs that can down regulate gene expression of target messenger RNA (mRNA) at the post-transcriptional level by partial or complete complimentary base pairing [9]. Abnormal expression of miRNAs has been well highlighted in various types of cancer, including breast cancer. Previously, two COX-2 upregulated miRNAs, miR526b and miR655, have been identified and established as oncogenic miRNAs in human breast cancer [10,11]. The roles of both miR526b and miR655 have been implicated in many hallmarks of cancer, including driving primary tumour growth, induction of stem-like cells (SLCs) phenotype, epithelial-to-mesenchymal transition (EMT), invasion and migration, as well as distant metastasis when tested in vivo in mouse models [10,11]. Moreover, both miRNAs target a transcription factor, *CPEB2A* gene, which was recently validated as a tumor suppressor [12]. However, the potential roles of miR526b and miR655 in tumour-associated angiogenesis and lymphangiogenesis in breast cancer have not yet been investigated. 

Angiogenesis and lymphangiogenesis are processes involving the formation of new blood or lymph vessels from pre-existing vasculature, both of which progress through the proliferation, migration, and maturation of nearby blood or lymph vessel endothelial cells [13,14]. Angiogenesis is an essential biological process that is fundamental for development, reproduction, and wound repair; however, this process is also considered a major hallmark of cancer [15]. As tumour growth can only reach 1–2 mm without sufficient blood supply, angiogenesis is essential during the uncontrolled growth of tumours for supply of sufficient oxygen and nutrients [15,16]. Similarly, lymphangiogenesis is an essential biological process that has also been implicated in many cancers, including breast cancer. The initial sites of metastasis in breast cancer are often the regional lymph nodes, and the migration of tumour cells to these sites is facilitated by lymphangiogenesis [17]. Vascular endothelial growth factors (VEGF) play a key role in both angiogenesis and lymphangiogenesis. Specifically, VEGFA is a major mediator of angiogenesis, through binding with VEGF receptors (VEGFR)1 and VEGFR2, leading to proangiogenic activity and migration of endothelial cells [18]. VEGFC and VEGFD are the primary ligands regulating lymphangiogenesis through binding of VEGFR3 on lymphatic endothelial cells; however, both ligands also have a weak affinity for VEGFR2, thus in part activating angiogenesis [18]. CD31 is also a well known stimulator of angiogenesis specifically involved in cell to cell interactions necessary for the organization of blood endothelial cells [19]. Moreover, lymphatic vessel endothelial hyaluronan receptor-1 (LYVE-1) [5,14] is a marker of lymphangiogenesis.

The majority of breast cancer patients are estrogen receptor (ER) positive and treated with tamoxifen [1]. Thus, in the present study, we investigated miR526b and miR655 and their potential roles in the process of breast cancer tumour-associated angiogenesis and lymphangiogenesis with an emphasis on ER-positive breast cancer, and we selected MCF7 as the breast cancer model to overexpress miRNAs. In vitro studies involving cell migration and capillary-like tube formation assays were conducted with human umbilical vein endothelial cells (HUVECs) and cell-free conditioned media collected from miR526b and miR655 overexpressing breast cancer cell lines [10,11]. Furthermore, the involvement of the COX2, *EP4*, and PI3K/Akt signalling pathways was investigated during angiogenesis assays by either COX-2 inhibition, *EP4* receptor antagonism, or PI3K/Akt pathway inhibition. We establish that overexpression of miR526b and miR655 is linked with tumour-associated angiogenesis and lymphangiogenesis in vitro. Our study also suggests that this stimulation occurs via the activation of the *EP4* receptor and subsequent activation of the PI3K/Akt pathway. In support of these findings, we used an in situ model to examine the relationship of miR526b and miR655 expression in human breast cancer tissues with expression of angiogenesis and lymphangiogenesis markers. Previously, we have shown that high expression of miR526b or miR655 in human breast cancer tissue is associated with reduced breast cancer patient survival [10,11]. Our present results demonstrate that both miR526b and miR655 expression is positively correlated with established angiogenesis and lymphangiogenesis markers in human breast cancer. Overall, our study establishes the roles of miR526b and miR655 in human breast cancer tumour-associated angiogenesis and lymphangiogenesis.

## 2. Results

### 2.1. Over-Expression of miR526b and miR655 in Poorly Metastatic (ER Positive) MCF7 Breast Cancer Cell Line Results in Upregulation of Angiogenesis and Lymphangiogenesis Markers

RNA was extracted from various passages of 90% confluent MCF7, MCF7-miR526b, and MCF7-miR655 cell lines and reverse transcribed into cDNA. The cDNA was then used to quantify the expression of known markers of lymphangiogenesis and angiogenesis. Relative gene expression fold change analysis was performed to compare the miRNA high cell lines, MCF7-miR526b and MCF7-miR655, to miRNA-low MCF7 cells. It was found that lymphangiogenesis marker, *VEGFD*, was significantly upregulated in MCF7-miR526b and MCF7-miR655 cell lines, while *VEGFC* and *LYVE-1* were marginally upregulated in MCF7-miR526b and significantly upregulated in MCF7-miR655 (Figure 1A). Furthermore, angiogenesis marker, *VEGFA*, was found to be significantly upregulated in both miRNA high cell lines (Figure 1A). Further, we extracted total protein from MCF7, MCF7-miR526b, and MCF7-miR655 cells for the quantification of VEGFA, VEGFC, VEGFD, COX-2, and LYVE-1 markers. MCF7-miR655 overexpressing cell line showed high expression of all VEGF markers and COX-2 expression at the protein level, while MCF7-miR526b showed significantly higher expression of VEGFC and VEGFD expression (Figure 1B,C). Whole Western blot data with corresponding molecular weights are presented in Appendix A.

Expression of receptors *VEGFR1* and *EP4* was found to be significantly upregulated in both the MCF7-miR526b and MCF7-miR655 cell lines (Figure 2A). *VEGFR2* was found to be marginally upregulated in MCF7-miR526b and MCF7-miR655 cell lines; however, this was not statistically significant (Figure 2A).

### 2.2. Human Umbilical Vein Endothelial Cells (HUVECs) Treated with MCF7-miR526b and MCF7-miR655 Conditioned Media Show Higher Expression of VEGF and EP4 Receptors

HUVECs grown to 90% confluency were treated overnight with conditioned media collected from MCF7-miR526b and MCF7-miR655 cell cultures. Total RNA was extracted from treated HUVECs and reverse transcribed to cDNA. TaqMan gene expression assay comparing the expression of *VEGFR1, VEGFR2,* and *EP4* was conducted. By virtue, HUVECs show very high expression of VEGFR1 and VEGFR2; here, we observed that miRNA-conditioned media further induced expression, showing a significant upregulation of *VEGFR2* in comparison with HUVECs treated with MCF7 conditioned media (Figure 2B). Moreover, expression of *VEGFR1* was found to be significantly upregulated in HUVECs treated with MCF7-miR526b conditioned media and marginally upregulated in those treated with MCF7-miR655 conditioned media (Figure 2B). HUVECs express a low level of *EP4* [20]; here, we observed a very significant upregulation of *EP4* expression in HUVECs treated with MCF7-miR655 conditioned media, and marginal upregulation following treatment with MCF7-miR526b conditioned media (Figure 2B).

### 2.3. Cancer Cell Conditioned Media Induces Migration and Tube Formation of HUVEC Cells

To examine the in vitro role of miR526b and miR655 in angiogenesis, we tested the cell migration capacity of HUVECs in cell-free conditioned media collected from MCF7, MCF7-miR526b, and MCF7-miR655 cell lines. Here, HUVEC cells were seeded and grown in a 24-well plate and a scratch wound was made with a 2 µL pipette tip. Cell-free conditioned from MCF7-miR526b and MCF7-miR655 cell lines were found to result in a significant increase in HUVEC migration during wound healing compared with the MCF7 conditioned media at the 24 h time point (Figure 3A,B). Cell migration images of the positive and negative controls, along with the experimental conditions at other time points, are provided in Appendix A.

To further investigate the effects of miR526b and miR655 on the angiogenesis potential of HUVECs, a tube formation assay was performed. The tube formation assay performed with growth factor reduced Matrigel is an established surrogate of angiogenesis in vitro. Cultured HUVECs were harvested and resuspended in MCF7, MCF7-miR526b, or MCF7-miR655 conditioned media, and seeded in a Matrigel-coated 24-well plate. Tube formation was observed and recorded at 0–24 h with image capturing. MCF7-miR655 conditioned media significantly stimulated an increase in the formation of both tubes and branching points at the 24 h time point when compared with the MCF7 conditioned media (Figure 4A–C), while MCF7-miR526b conditioned media resulted in a marginal increase of tubes, and a very significant increase in branching points at the 24 h time-point (Figure 4A–C). Tube formation images of the positive and negative controls, along with the experimental conditions at other time points, are provided in Appendix A.

### 2.4. Treatments with COX2 Inhibitor (COX-I) and EP4 Antagonist (EP4A) Significantly Inhibits miRNA Induced Functions

PGE2 is the major product of COX-2 enzyme activity. The activation of *EP4* receptor by binding of PGE2 results in activation of the PI3K/Akt signalling pathway and is associated with promotion of tumour cell migration and angiogenesis [6]. To examine whether the stimulatory actions of MCF7-miR526b and MCF7-miR655 is the result of involvement of COX-2 activity or *EP4*-signalling, we tested both the migration (Figure 3) and tube formation (Figure 5) phenotypes of HUVECs stimulated with miRNA-conditioned media, with the addition of either COX-2 inhibitor (COX2-I, NS398), *EP4* antagonist (*EP4A*, ONO-AE3-208), or PI3K/Akt pathway inhibitor (Wortmannin, WM).

#### 2.4.1. Inhibition of Cell Migration

Specifically, for MCF7-miR526b conditioned media, the addition of COX2-I or *EP4A* significantly inhibited migration of HUVECs at the 8 h and 24 h time points, when compared with the vehicle. (images at 24 h are shown in Figure 3C, and quantification in Figure 3D). Moreover, for MCF7-miR655 conditioned media, addition of COX2-I and *EP4A* significantly inhibited HUVEC migration at 24 h (images at 24 h are shown in Figure 3E and quantification provided in Figure 3F). Additional time points of migration of HUVECs with COX2-I or *EP4A* are presented in Appendix A.

#### 2.4.2. Inhibition of Tube Formation

The addition of COX2-I, *EP4A*, or WM to MCF7-miR655 conditioned media significantly reduced the number of both tubes and branching points formed by HUVECs at the 24 h timepoint (images at 24 h are shown in Figure 5A and quantification provided in Figure 5B,C). While COX2-I could inhibit miR655-conditioned media induced tube formation and branching formation, the addition of either *EP4A* or WM resulted in complete inhibition tube formation of HUVECs as early as 2 h, and we observed the same trend in all other time points compared with the vehicle. Quantitative data are presented for 2 h, 12 h, and 24 h (Figure 5B,C), and images are presented only at 24 h (Figure 5A). Images captured at other time points are provided in Appendix A.

### 2.5. Expression of miR526b and miR655 in Human Breast Tumour Tissue Correlated with Angiogenesis and Lymphangiogenesis Markers

We investigated miRNA expression *in situ* with human breast tumour tissue retrieved from Ontario Institute of Cancer Research (OICR) Tumour Bank to examine the relationship of miR526b and miR655 expression with expression of angiogenesis and lymphangiogenesis markers. We used 105 tumour tissue samples and 20 non-cancerous tissues. Demographic data of the sample are provided in Table 1 and a description of the sample has been published previously [8,10,11,21]. The majority of the tumor samples used in this study were ER and PR positive and HER-2 negative; with only 10 triple negative breast cancer samples. Moreover, this set has no stage IV tumor and only a few stage I tumor samples. Taqman gene expression (quantitative real time polymerase chain reaction; qRT-PCR) analysis was performed on cDNA produced from each sample and the relative fold change of mRNA was measured to compare control and tumour tissues. We found that the tumour samples illustrated higher expression of *VEGFA* (29.2 fold) than the control group (Figure 6A). The expression of both miR526b and miR655 in this sample set was quantified and published earlier [10,11]. Here, we evaluated the correlation between miRNA expression and *VEGFA* expression in tumour samples. Both miRNAs show a positive correlation with *VEGFA*, with miR526b having a correlation coefficient of R = 0.338 (*p* < 0.001) (Figure 6B), and miR655 having a correlation coefficient of R = 0.425 (*p* < 0.0001) (Figure 6C). Therefore, in both cases, *VEGFA* expression increases as miRNA expression increased. We previously reported the expression of angiogenesis marker *CD31* in this same sample set [21]. Here, we show that the expression of both miR526b and miR655 was significantly correlated with *CD31* expression, showing a positive correlation of R = 0.463 (*p* < 0.0001) for miR655 (Figure 6D) and R = 0.526 (*p* < 0.0001) for miR526b (Figure 6E).

We have previously measured lymphangiogenesis markers *VEGFC* and *LYVE-1* expression in these tumour samples [21]. For the first time here, we report that *VEGFD* expression is significantly high in tumour samples compared with control tissue (Figure 7A). Expression of miR526b and miR655 in human tumour tissue was significantly correlated with lymphangiogenesis markers *VEGFC*, *VEGFD,* and *LYVE-1* (Figure 7). Specifically, miR526b and miR655 were very significantly correlated with *VEGFD*. Not only did *VEGFD* show significantly higher expression in tumour samples (Figure 7A), a strong positive correlation of R = 0.7652 (*p* < 0.00001) with miR526b (Figure 7B) and R = 0.933 (*p* < 0.00001) with miR655 (Figure 7C) is observed. Furthermore, miR526b had a positive correlation coefficient of R = 0.5286 (*p* < 0.00001) with *VEGFC* (Figure 7D), and miR655 had a coefficient of R = 0.6053 (*p* < 0.00001) with *VEGFC* (Figure 7E). Expression of *LYVE-1* was also positively correlated with both miRNAs, showing a positive correlation coefficient (R = 0.5256, *p* < 0.00001) for miR526b in Figure 7F as well as for miR655 (Figure 7G) (R = 0.3437, *p* = 0.001279).

Further, we investigated if there is any difference in the distribution of high miRNA expression across hormone status (Table 2). We subdivided miRNA expression according to low (+ΔCt value) and very high miRNA expression (−ΔCt value) in various tumor stages and different hormone receptor status. Both miRNA high samples were higher in proportion in ER and PR positive and HER2 negative samples; however, none of these distributions were statistically significant.

## 3. Discussion

In this study, the role of miR526b and miR655 in breast tumour-associated angiogenesis and lymphangiogenesis was investigated. The roles of specific miRNA in tumor-associated angiogenesis and lymphangiogenesis have been well highlighted. In a study by Cascico et al., oncogenic miR-20b was shown to be involved in the regulation of VEGF in breast cancer cells by targeting HIF-1α [22], while the role of miR-10b as an angiogenic regulator was validated in a study by Liu et al. [23]. Expression of VEGFs has been found to be associated with other miRNAs involved in the regulation of angiogenesis and lymphangiogenesis, such as miR-182 [24] and miR-20a [25], which were both found to cause an upregulation of VEGFA; in contrast, another study has highlighted the role of miR-128 as a tumour suppressor miRNA, which results in the reduction of both VEGFA and VEGFC expression [26]. Further, other studies have shown that down regulation of tumour suppressive miR-126 [27] and miR-128 [26] results in the promotion of tumour-associated lymphangiogenesis, which was assessed by the subsequent reduction of a lymphangiogenic marker LYVE1. Therefore, miRNA can directly or indirectly regulate both angiogenesis and lymphangiogenesis by regulating expression of angiogenesis and lymphangiogenesis markers.

We have previously shown that COX-2 overexpression in ER-positive MCF7 breast cancer cells significantly increased expression of miR526b and miR655 [10,11]. Interestingly, overexpression of these two miRNAs in poorly metastatic and ER positive MCF7 cells promotes aggressive breast cancer phenotypes [10,11]. Recently, a common target of both miRNAs *CPEB2* has been validated as a tumor suppressor gene in breast cancer [12]. However, there was no direct report on these two miRNAs and their roles in breast cancer-associated angiogenesis and lymphangiogenesis. miR526b and miR655 can upregulate COX-2 expression via the NFKB pathway [10,11]. It is very well known that COX-2 induces angiogenesis and lymphangiogenesis through production of VEGFC and VEGFD, and upregulation of PI3K/Akt signalling and COX-2 overexpression can also induce LYVE-1 over expression in mouse breast tumours [28,29]. We have previously shown low expression of miR526b and miR655 in ER receptor positive cell lines MCF7 and T47D, and that these miRNA have the highest expression in triple negative breast cancer cells MDAMB231, MCF7-COX2, and Hs578T, with a relative correlation of miRNA expression with COX-2 expression [11]. Thus, we chose to overexpress miR526b and miR655 in an ER positive, poorly metastatic breast cancer cell MCF7 to investigate miRNA gain of functions.

In this study, we show that miR526b and miR655 overexpression results in significant upregulation of angiogenesis and lymphangiogenesis markers, specifically, VEGFA, VEGFC, VEGFD, COX-2, and LYVE-1. Furthermore, the expression of angiogenesis and lymphangiogenesis receptors in miRNA high cells was measured to test the autocrine regulation of miRNA in tumor associated angiogenesis. Although VEGFR1 and VEGFR2 receptors are primarily expressed on endothelial cells, previous studies have reported VEGFR expression in breast cancer cells, establishing the involvement of a VEGF–VEGFR autocrine loop [30]. VEGFR1 expression in breast cancer cells might promote tumour growth and metastasis, and has been established as an unfavourable indicator of progression in breast cancer patients [31]. Moreover, a VEGFR2 autocrine signalling loop has been established in breast cancer cell lines, and has been shown to activate MAP kinase pathways [32]. Here, we show that miR526b and miR655 overexpression in MCF7 cell line results in an extremely significant upregulation of *VEGFR1* expression at the mRNA level. *VEGFR2* expression was also found to be marginally upregulated at the mRNA level in MCF7-miR526b and MCF7-miR655. These results suggest that the upregulation of VEGFA, VEGFC, and VEGFD in MCF7-miR526b and MCF7-miR655 results in the production and release of these ligands into the tumour microenvironment, which may feed into an autocrine loop on breast cancer cells to activate VEGFR1 and VEGFR2 expression. Furthermore, it has been previously shown that high expression of COX-2, and thus overproduction of PGE2, leads to overexpression of miR526b and miR655 in breast cancer [10,11]. Production of PGE2 by COX-2 results in upregulation of PGE2 receptors EPs. Specifically, binding of PGE2 to receptor *EP4* stimulates non-canonical pathways PI3K/Akt and ERK [6], and is associated with tumour-associated angiogenesis and lymphangiogenesis [6,7,14,33,34]. Here, expression of *EP4* receptor was found to be significantly upregulated in MCF7-miR526b and MCF7-miR655 cells, suggesting that PGE2 released into the tumour microenvironment is involved in autocrine signalling through *EP4* receptor on breast cancer cells.

In this study, we also tested the paracrine potential of miR526b and miR655. Other reports have demonstrated the effects of miRNA-overexpressing cell lines and their involvement with angiogenesis in vitro through secretions of stimulatory proteins or by co-culture with endothelial cells such as miR-155 [35], miR-494 [36], and miR-182 [24], which were all established to promote endothelial tube formation and migration. Similarly, we investigated the paracrine potential of MCF7-miR526b and MCF7-miR655 cell line secretions and tested the effects on HUVEC cell tube formation and migration potential. Here, we observed that the conditioned media collected from miR526b and miR655 overexpressing breast cancer cell lines stimulates both tube formation and migration of endothelial cells. Moreover, we found *VEGFR1* and *VEGFR2* to be upregulated in HUVEC cells following treatment with miRNA overexpressing cell conditioned media, and that expression of *EP4* receptor was found to be upregulated in HUVECs treated with miRNA high cell line conditioned media. Because *EP4* is a main receptor that regulates COX-2/PGE2 induced functions [8,10,11,28,29,34], this led us to further investigate the involvement of this *EP4* signalling pathway in miRNA induced angiogenesis and lymphangiogenesis. To test *EP4* signalling mechanisms, we used a specific COX-2I, *EP4A* or PI3K/Akt inhibitor along with miRNA overexpressing cell conditioned media. *EP4A* could significantly abrogate HUVECs’ migration and tube formation, however, the PI3K/Akt inhibitor completely blocked these phenotypes, suggesting that miR526b and miR655 induce angiogenesis and lymphangiogenesis via *EP4*/PI3K/Akt pathways. Thus, miR526b and miR655 regulate angiogenesis through the production of VEGFs and their subsequent release into the tumour microenvironment for both paracrine and autocrine signalling.

To further investigate the translational involvement of miR526b and miR655 in tumour associated angiogenesis and lymphangiogenesis, we examined the expression of angiogenesis and lymphangiogenesis markers in situ. The majority of the tumor samples used in this study are ER and PR positive and HER2 negative, with only a few triple negative breast cancer samples. We have previously shown that in this set of human tumour samples both miR526b and miR655 expression is high [10,11]. We found proportionately more samples with high miRNA expression in the ER positive samples, but the distribution was not statistically significantly, which could be because of the fact that we have only a few sample with very high expression. We have also previously shown expression of *CD31, VEGFC*, and *LYVE1* to be high in this tumour set [21]. Here, we show that in malignant breast samples, expression of *VEGFA* and *VEGFD* is significantly high compared with the disease-free control samples. Moreover, we show that expression of both miR526b and miR655 is positively correlated with expression of angiogenesis (*CD31* and *VEGFA*) and lymphangiogenesis markers (*VEGFC, VEGFD,* and *LYVE1*) in human breast cancer tissue. We found the strongest correlation of miRNA expression with *VEGFD* expression in tumor samples, which confirms our cell line findings of high expression of VEGFD in miRNA high cell lines. This observation is also supported by other studies showing a correlation of miRNA with angiogenic markers, including the positive correlation of miR-20a expression in breast tumour tissue with *VEGFA* expression [25]. Further, a study by He et al. found that tumour suppressive miR-186 was negatively correlated with expression of *VEGFC* in tumour tissue collected from bladder cancer patients [37].

The exact mechanisms by which miR526b and miR655 stimulate angiogenesis and lymphangiogenesis remain unknown. However, we have shown that COX-2 overexpression results in overexpression of both miR526b and miR655, and further results in overproduction of inflammatory PGE2. We have also shown that miRNA overexpression induces COX-2 expression via NFKB pathway [10,11]. Therefore, miRNA overexpressing cell lines are high in both production of COX-2 and secretions of PGE2. Furthermore, with bioinformatics analysis, we have previously reported that *PTEN* (phosphatase and tensin homolog) is a target of both miR526b and miR655 [10]. Here, we validated that *PTEN* is indeed a direct target of miR526b and miR655, as it is significantly down regulated in both MCF7-miR526b and MCF7-miR655 cell lines compared with MCF7 (Figure 8A). *PTEN* is also a known target of other established miRNAs, such as miR-494, which has been reported by Mao et al. to target *PTEN* in response to hypoxic conditions to promote angiogenesis and tumour growth in non-small lung cancer [36]. *PTEN* acts as a tumour suppressor gene both by inhibiting the PI3K/Akt pathway, and by acting as a negative regulator of HIF-1α (hypoxia-inducible factor-1) [38,39]. In turn, HIF-1α is a transcription factor known to promote the transcription of many angiogenesis-associated genes, including VEGFs [40]. Although we did not measure HIF-1α expression in our samples, we speculate that both miR526b and miR655 target this pathway, resulting in VEGF secretions into the tumour microenvironment, thus facilitating tumour-associated angiogenesis and lymphangiogenesis. The proposed pathway is presented in Figure 8B. A similar report in triple negative breast cancer was shown, in which *PTEN* downregulation resulting in cell proliferation via the PI3K pathway was shown [41].

Here, we established the roles of miR526b and miR655 as promoter and regulator of breast cancer angiogenesis and lymphangiogenesis using an ER positive breast cancer cell model MCF7 and showed that miRNA expression is high in ER positive breast cancer samples and highly correlated with angiogenesis and lymphangiogenesis markers in breast cancer. Further investigation into the roles of these miRNAs in the triple negative breast cancer model and incorporation of more tumor samples of various hormonal receptor status and tumor stages would give greater insight into the mechanisms of these miRNA across differential subtypes of breast cancer.

## 4. Materials and Methods

### 4.1. Ethics Statements

Brandon University Research Ethics Committee approves this study (#21986, 21 April 2017). The human tissues used in this project were obtained by Dr. Peeyush K Lala at the University of Western Ontario (UWO) from the Ontario Institute for Cancer Research (OICR) repository (created on the basis of donor consent) following approval of human ethics by the Ethics Review Board of the OICR and UWO. Total RNA and miRNA were extracted using Qiagen (Qiagen, Toronto, ON, Canada) RNA and miRNA extraction kits followed by cDNA synthesis at UWO, and a portion of cDNA of all samples were transferred to Dr. Mousumi Majumder at Brandon University following a material transfer agreement.

### 4.2. Cell Culture

Human breast cancer cell line MCF7 was purchased from American Type Culture Collection (ATCC, Rockville, MD, USA). Stable miRNA overexpressing MCF7-miR526b and MCF7-miR655 cell lines were established as previously described [9,10]. MCF7, MCF7-miR526b, and MCF7-miR655 cells were all grown in minimal essential medium (MEM) (Life Technologies, Thermofisher, Ottawa, ON, Canada) supplemented with 10% fetal bovine serum (FBS) and 1% Penstrep. Furthermore, MCF7-miR526b and MCF7-miR655 cell lines were sustained with Geneticin (Life Technologies Thermofisher, Ottawa, ON, Canada) at 200 ng/mL.

HUVECs were purchased from Life Technologies and grown in Medium 200 (GIBCO, ON) supplemented with low serum growth supplement (LSGS) kit (GIBCO, Toronto, ON, Canada) containing 2% FBS, hydrocortisone (1 µg/mL), human epidermal growth factor (10 ng/mL), basic fibroblast growth factor (3 ng/mL), and heparin (10 µg/mL). All cell lines were maintained in a humidified incubator at 37 °C with 5% CO_2_.

### 4.3. Collection of Conditioned Media

MCF7, MCF7-miR526b, and MCF7-miR655 cell lines were grown in complete serum supplemented media until 90% confluent. Cells were then washed with 1X phosphate buffered saline (PBS) to remove any trace of serum. The cells were then starved with basal MEM (serum-free) for 12 h prior to collection of media. Cell free supernatant was then collected for HUVEC functional assays.

### 4.4. Drugs and Chemicals

NS398 (COX-2 inhibitor) was purchased from Cayman Chemical (Ann Arbor, MI, USA). ONO-AE3-208 (selective *EP4* antagonist, *EP4A*) was a gift from ONO Pharmaceuticals, Osaka, Japan. Wortmannin (WM), an irreversible PI3K inhibitor purchased from Sigma-Aldrich (Saint Louis, MO, USA), Dr. Lala generously shared 1 mM of each drug with us. For all treatments in vitro, DMSO (vehicle) served as the control.

### 4.5. Tube Formation Assay

The assay was carried out as previously described [21,34], using a 24-well plate. HUVECs were resuspended in either non-supplemented, serum free Medium 200 to serve as a negative control; MCF7 conditioned media, MCF7-miR526b conditioned media, or MCF7-miR655 conditioned media as experimental conditions; or complete serum supplemented Medium 200 to serve as the positive control. Each condition was tested in triplicate (experimental replicates) and repeated three times (biological replicates). HUVECs with each condition were then seeded on a growth factor reduced Matrigel (BD Biosciences, Mississauga, ON, Canada) coated 24-well plate. Matrigel was prepared using a 1:2 ratio with one part Matrigel and two parts un-supplemented Medium 200. Tube formation was examined at different time intervals, and images were obtained with a Nikon inverted microscope. Quantification of tubes and branching points was carried out using NIH Image J software (Version 64, NIH, Bethesda, MD, USA). To test the involvement of COX-2, *EP4* receptor, and the PI3K/Akt pathway, tube formation assay was repeated with cancer cell conditioned media along with 20 µM NS398 or 50 µM ONO-AE3-208. To confirm the involvement of the PI3K/Akt signalling pathway, we also used 10 µM WM, an irreversible P13K/Akt inhibitor.

### 4.6. HUVEC Migration Assay

HUVECs were grown in LSGS supplemented Medium 200 in a T75 flask, then harvested and re-suspended in supplemented Medium 200, after which 300 μL of harvested cells was added in a 24-well attachment plate and maintained until 90% confluency. The surface of each well was scratched with a 2 µL sterile micropipette tip and cells were washed with PBS. Each condition was then applied to the wells. Basal Medium 200 served as the negative control, and serum supplemented Medium 200 acted as the positive control. MCF7, MCF7-miR526b, and MCF7-miR655 conditioned media were the experimental conditions. A total of 300 µL of the respective condition was added per well. To test the involvement of the *EP4* and the PI3K/Akt signalling pathway, 300 µL of cell-free conditioned media from MCF7-miR526b or MCF7-miR655 was added to the wells. In another set of experiments along with cancer cell conditioned media, additionally either 20 µM NS398, 50 µM ONO-AE3-208, or 10 µM WM was added, and DMSO served as a control to test the involvement of COX-2, *EP4*, and the PI3K/Akt signalling pathway in cell migration. The progress of HUVEC migration and photos of the scratch wound size were captured using an inverted microscope at differing time intervals, and NIH ImageJ software was used to measure the width of the wound in pixels.

### 4.7. Quantitative Real-Time PCR (qRT-PCR)

Total RNA was extracted from MCF7, MCF7-miR526b, and MCF7-miR655 cell lines using miRNeasy Mini Kit (Qiagen, Toronto, ON, Canada) and reverse transcribed using the TaqMan microRNA and mRNA cDNA Reverse Transcription Kit (Applied Biosystems, Waltham, MA, USA). The TaqMan MiRNA or gene expression assays was used for quantitative PCR. Two control markers, *Beta-actin* (Hs01060665_g1) and *RPL5* (Hs03044958_g1) expression was quantified using RT-PCR and used to normalize the expression of the following angiogenesis and lymphangiogenesis markers using relative analysis: *VEGFA* (Hs00900055_m1), *VEGFC* (Hs01099203_m1), *FIGF* (Hs01128659_m1), *PTGS2* (Hs00153133_m1), *LYVE1* (Hs00272659_m1), *CD31* (Hs01065279_m1), and *PTGER4* (Hs00168761_m1). Moreover, expression of tumour suppressor gene *PTEN* (Hs0082981_s1) was examined. Gene expression was measured using delta CT values to obtain the fold change, as described earlier [42].

For quantification of receptor expression on HUVECs, HUVECs were grown in a 6-well plate until confluent. HUVECs were then treated with MCF7, MCF7-miR526b, or MCF7-miR655 conditioned media for 12 h. HUVECs were then trypsinized and collected for RNA extraction and cDNA synthesis. Quantitative qPCR was carried out with *FLT1* (Hs01052961_m1) (*VEGFR1*)*, KDR* (Hs00964383_g1) (*VEGFR2),* and *PTGER4* (Hs00911700) (*EP4*) as described above. We also measured *VEGFR1, VEGFR2,* and *EP4* in MCF7, MCF7-miR526b, and MCF7-miR655 cell lines.

### 4.8. Western Blot

Cancer cells were treated with M-PER^®^ Mammalian Protein Extraction Reagent (Thermo Scientific, Rockford, IL, USA), HALT Protease Inhibitor Cocktail (Thermo Scientific), and Phosphatase Inhibitor Cocktail (Thermo Scientific) to extract total protein. A total of 15–20 µg of total protein were electrophoresed per well on a SDS-polyacrylamide gel; transferred onto Immobilon-FL PVDF membranes (Millipore, Billerica, MA, USA); and further incubated with the following primary antibodies: VEGFA (sc-507), VEGFC (sc-1881), VEGFD (sc-13085), LYVE1 (sc-28190), and COX-2 (sc-1747), using antibodies (1:500 dilutions) from Santa Cruz Biotechnology (Santa Cruz Biotechnology, Santa Cruz, CA, USA). Monoclonal GAPDH antibody (MAB374) was from Millipore, Billerica, MA, USA. After blocking with primary antibodies, overnight blots were probed with a mixture of IRDye polyclonal secondary antibodies (LI-COR Biosciences, Lincoln, NE, USA). Images were read with an Odyssey infrared imaging system (LI-COR Biosciences).

### 4.9. Human Breast Cancer Tissue Samples

Frozen human breast tumour (*n* = 105) and control (*n* = 20) tissue samples were obtained previously from the Ontario Tumour Bank with the demographic description shown in Table 1. Qiagen miRNeasy mini kit was used to extract mRNA or miRNA from tissue samples, followed by cDNA synthesis using cDNA Reverse Transcription Kit (Life Technologies, Applied Biosystems, cat # 4368814, ON, Canada) To examine the potential correlations between miRNA and lymphangiogenesis or angiogenesis markers, tissue sample cDNA was screened using qRT-PCR. Expression of *miR526b* (Hs03304873_pri) and *miR655* (Hs03296227_pri); angiogenesis markers *VEGFA* (Hs00900055_m1) and *CD31* (Hs01065279_m1); and lymphangiogenesis markers *LYVE1* (Hs00272659_m1), *VEGFC* (Hs01099203_m1), and *VEGFD* (Hs01128659_m1) were all examined.

### 4.10. Statistical Analysis

Statistical calculations were performed using GraphPad Prism software version 5 (GraphPad Software, San Diego, CA, USA). All parametric data were analyzed with one-way analysis of variance (ANOVA) followed by Tukey–Kramer or Dunnett post-hoc comparisons. Student’s *t*-test was used when comparing two datasets and Pearson’s coefficient was employed to assess statistical correlations. We used Z-score to compare the proportion of miRNA high expression in various tumor grades and ER, PR, and HER2 status positive and negative samples in the tumors. Statistically relevant differences between means were accepted at *p* < 0.05.

## 5. Conclusions

The roles of angiogenesis and lymphangiogenesis in breast cancer tumour growth and metastasis have been well established. Breast cancer metastasis requires that primary tumour cells possess the ability to enter the blood or lymph vessels and travel to secondary sites [43]. This is greatly facilitated by angiogenesis and lymphangiogenesis [44,45]. Overall, our study establishes the involvement of microRNA (miR526b and miR655) in tumour-associated angiogenesis and lymphangiogenesis, specifically in ER positive breast cancer. These findings further establish the involvement of these miRNAs in breast cancer metastasis and their potential as future breast cancer biomarkers. Our study also validates the involvement of PGE2 signalling through *EP4* receptor, and the subsequent PI3K/Akt pathways in these processes, further validating the use of *EP4* receptor antagonists as a potential therapeutic target in COX2-high breast cancer patients.

## Figures and Tables

**Figure 1 cancers-11-00938-f001:**
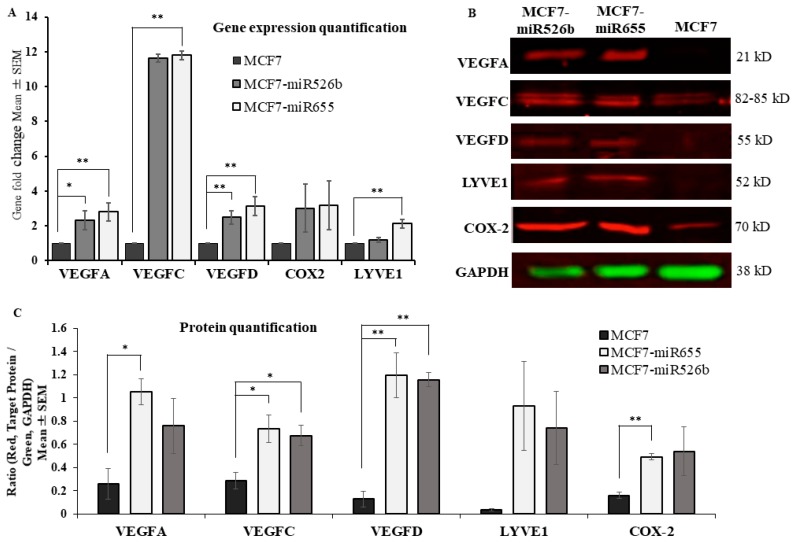
Overexpression of miR526b and miR655 in MCF7 cell line results in upregulation of angiogenesis and lymphangiogenesis markers. (**A**) Quantitative real time polymerase chain reaction (qRT-PCR) shows angiogenic and lymphangiogenic markers at the mRNA level with significant positive fold changes in the miRNA overexpressed cell lines, MCF7-miR526b and MCF7-miR655, compared with the MCF7 cell line. (**B**) The Western blot analysis shows a larger presence of angiogenesis and lymphangiogenesis markers at the protein level in the MCF7-miR655 and MCF7-miR526b cell lines compared with the MCF7 cell line. Green: control marker Glyceraldehyde 3-phosphate dehydrogenase (GAPDH), red: target proteins, either VEGFA or VEGFC or VEGFD or LYVE1 or COX-2. (**C**) Quantitative analysis of Western blot shows increased levels of angiogenesis and lymphangiogenesis markers. RT-PCR and Western blot quantitative data are presented as the mean ± SEM of triplicate replicates; * *p* < 0.05, ** *p* < 0.01. LYVE-1—lymphatic vessel endothelial hyaluronan receptor-1; VEGF—vascular endothelial growth factors; COX-2—cyclo-oxygenase 2.

**Figure 2 cancers-11-00938-f002:**
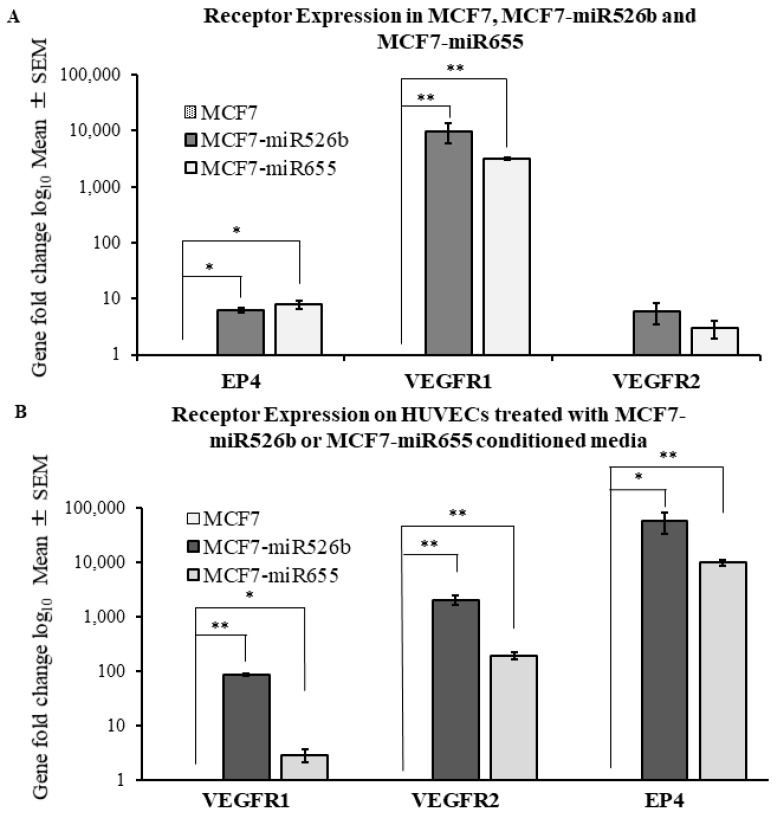
Overexpression of miR526b and miR655 in MCF7 or treatment of human umbilical vein endothelial cells (HUVECs) with miRNA-conditioned media results in upregulation of angiogenesis and lymphangiogenesis receptor markers. (**A**) Quantitative RT-PCR analysis shows significant upregulation of *prostaglandin E2 receptor 4 (EP4)*, *VEGF receptor 1 (VEGFR1)*, and *VEGFR2* expression in the MCF7-miR526b and MCF7-miR655 cell lines, when compared with MCF7. (**B**) Quantitative RT-PCR analysis shows the HUVECs treated with MCF7-miR526b or MCF7-miR655 conditioned media have greater receptor gene expression compared with HUVECs treated with MCF7 conditioned media. RT-PCR quantitative data are presented as the mean ± SEM of triplicate replicates; * *p* < 0.05, ** *p* < 0.01.

**Figure 3 cancers-11-00938-f003:**
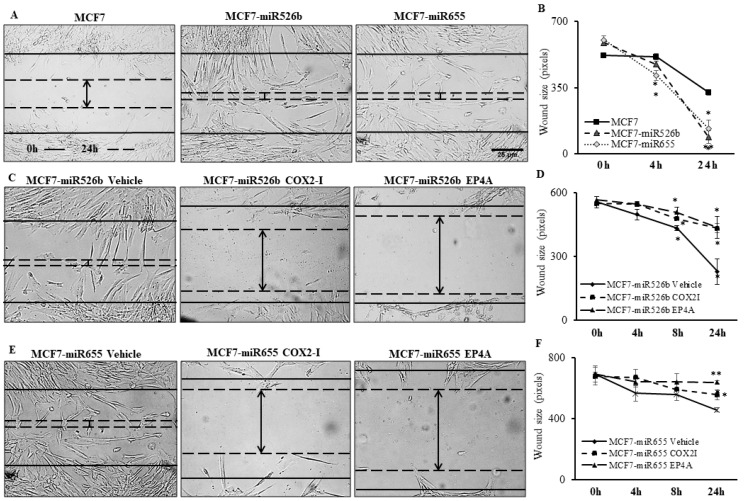
MCF7-miR526b and MCF7-miR655 conditioned media promotes cellular migration of HUVECs and inhibition of the COX2/*EP4* signaling pathway abrogates these phenotypes. Baseline scratches represented by black line; wound size at 24 h represented by dashed line. (**A**) Images of migration assay with conditioned media from MCF7, MCF7-miR526b, and MCF7-miR655 at the 24 h time point. (**B**) Quantitative data representing wound size per time point in conditioned media from MCF7, MCF7-miR526b, or MCF7-miR655. (**C**) Images of migration assay with conditioned media from MCF7-miR526b with the addition of vehicle, COX2-I, or EP4A, at the 24 h time point. (**D**) Quantitative data representing wound size per time point in conditioned media from MCF7-miR526b with the addition of either vehicle, COX2-I, or *EP4*A. (**E**) Images of migration assay with conditioned media from MCF7-miR655 with the addition of vehicle, COX2-I, or *EP4A*, at the 24 h time point. (**F**) Quantitative data representing wound size per time point in conditioned media from MCF7-miR655 with the addition of either vehicle, COX2-I, or *EP4A*. Data shown as mean ± SEM of three biological replicates, including three experimental replicates per biological replicate; * *p* < 0.05, ** *p*< 0.01.

**Figure 4 cancers-11-00938-f004:**
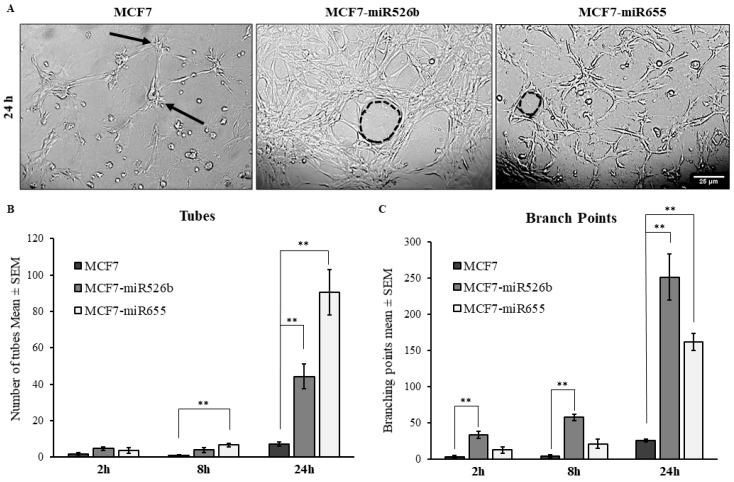
Overexpression of miR526b and miR655 results in an increase of tube formation of HUVECs. (**A**) Visual representation of tube formation of HUVECs at the 24 h time point in conditioned media from MCF7, MCF7-miR526b, or MCF7-miR655 cell line. Both tubes (dotted circle) and branching points (arrows) were greater in MCF7-miR526b or MCF7-miR655 conditioned media compared with MCF7 conditioned media. (**B**) Quantitative data represent number of tubes per time point, per condition. (**C**) Quantitative data represent number of branching points per time point, per condition. Quantitative data presented as the mean ± SEM of three biological replicates, including three experimental replicates per biological replicate; ** *p* < 0.01.

**Figure 5 cancers-11-00938-f005:**
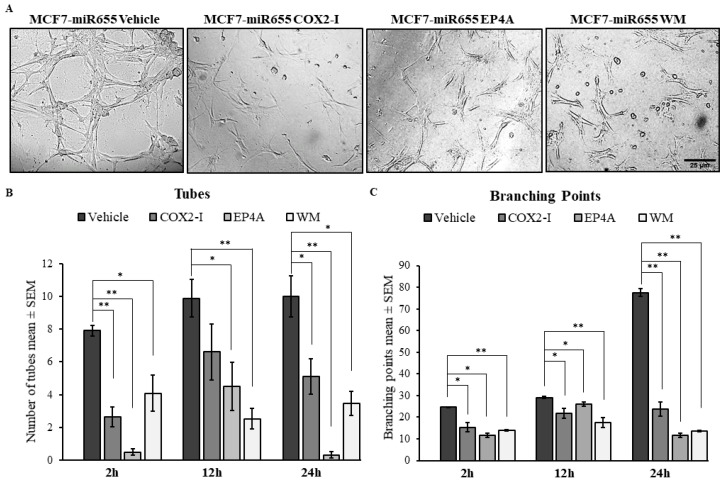
Treatment with COX2-I, *EP4A,* or PI3K pathway inhibitor (Wortmannin, WM) abrogates the tube formation stimulation abilities of MCF7-miR655 conditioned media. (**A**) Images of all treatments; miRNA-conditioned media with vehicle, COX2-I, *EP4A*, or PI3K inhibitor (WM) conditions at the 24 h time point. (**B**) Quantitative analysis of number of tubes formed per time point, per condition. (**C**) Quantitative analysis of branching points per time point, per condition. Quantitative data presented as the mean ± SEM of three biological replicates, including three experimental replicates per biological replicate; * *p* < 0.05, ** *p* < 0.01.

**Figure 6 cancers-11-00938-f006:**
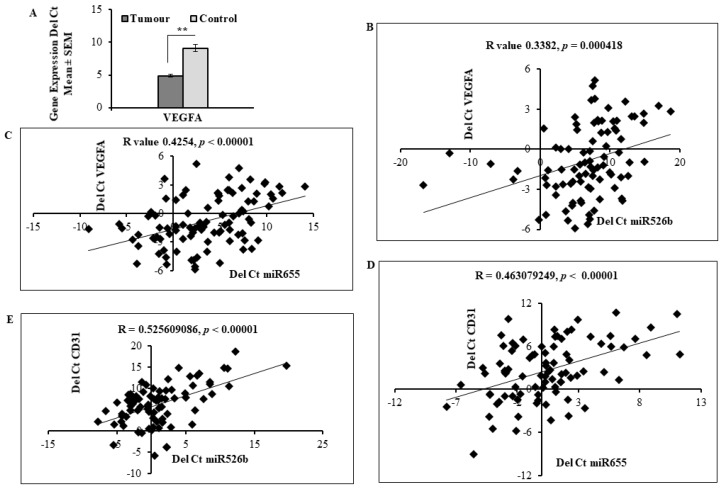
miR526b and miR655 expression is positively correlated with angiogenesis and vascular markers in human breast tumours. (**A**) qRT-PCR analysis of angiogenesis marker *VEGFA* mRNA expression in control (adjacent non-tumour) and tumoural tissues. Data are represented as mean ± SD. (**) indicates significant differences (*p* < 0.01). (**B**,**C**) miR526b and miR655 expression is positively correlated with *VEGFA* in primary breast cancer samples; Pearson’s coefficient indicates positive, but moderate correlations. (**D**,**E**) miR526b and miR655 expression level is positively correlated with *CD31* in primary breast cancer samples. Pearson’s coefficient suggests moderate correlation between the two variables.

**Figure 7 cancers-11-00938-f007:**
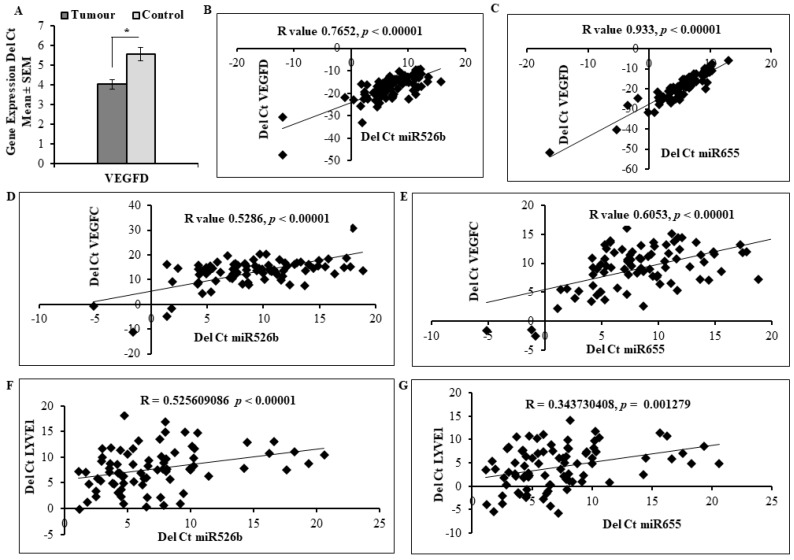
miR526b and miR655 expression is positively correlated with lymphangiogenesis markers in human breast tumours. (**A**) qRT-PCR analysis of *VEGFD* mRNA expression in control (adjacent non-tumour) and tumoural tissues. Data are represented as a mean ± SD. (*) indicates significant differences (*p* < 0.05). (**B**,**C**) miR526b and miR655 expression is positively correlated with the *VEGFD* in primary breast cancer samples; Pearson’s coefficient indicates a very strong positive correlation. (**D**,**E**) miR526b and miR655 expression level is positively correlated with another lymphangiogenesis marker *VEGFC* in primary breast cancer samples. Pearson’s coefficient suggests a positive correlation between the two variables. (**F**,**G**) Finally, *LYVE1* mRNA expression is positively correlated with miR526b and miR655 expression in tumour samples. Pearson’s coefficient suggests a moderate correlation between the two variables.

**Figure 8 cancers-11-00938-f008:**
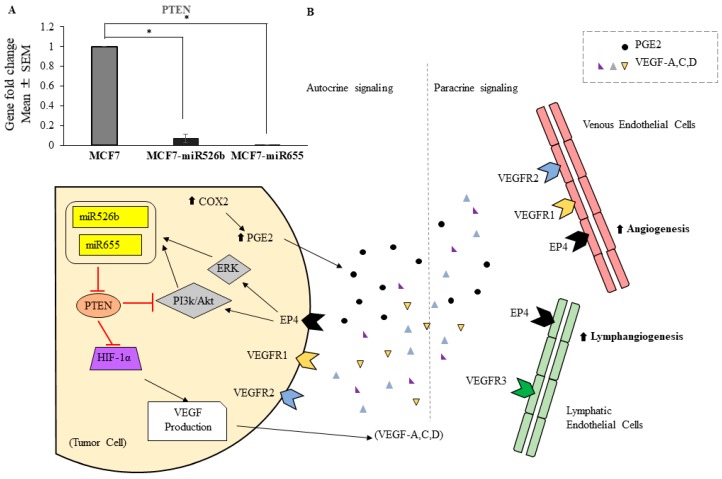
(**A**) miR526b and miR655 overexpressing cells showing significant down regulation of tumour suppressor gene *phosphatase and tensin homolog (PTEN)* mRNA expression. Data are represented as mean ± SD.; * (*p* < 0.05). (**B**) The proposed pathway of tumour associated angiogenesis and lymphangiogenesis promotion by miR655 and miR526b. miRNA over expressing cells are high in COX-2 expression, resulting in production of PGE2 and its subsequent release into the tumour microenvironment. PGE2 can signal through *EP4* in both an autocrine and paracrine fashion. Signalling through *EP4* on tumour cells stimulates ERK and PI3k/Akt signalling pathways, resulting in the upregulation of miR526b and miR655. Subsequently, miR526b and miR655 target tumour suppressor gene *PTEN*. Thus, the inhibitory effect of *PTEN* on hypoxia-inducible factor-1 (HIF-1a), as well as the PI3K/Akt pathway, is prevented by upregulation of these miRNAs. Without regulation, transcription factor HIF-1a promotes VEGFA, C, D transcription, and thus an overproduction of VEGFs and their subsequent release into the tumour microenvironment for both autocrine and paracrine signalling. VEGF molecules will bind to various VEGF receptor (VEGFR1, VEGFR2) molecules present on vascular endothelial cells (VECs) and lymphatic endothelial cells (LECs) to promote angiogenesis and lymphangiogenesis, respectively. In the absence *PTEN*’s regulation on PI3k/Akt pathway within the tumour cell, growth and proliferation is promoted.

**Table 1 cancers-11-00938-t001:** Demography, tobacco exposures, tumour grade, and hormone status of the control and tumor samples used in this study. Patient demography of human tissue biopsy samples illustrated. Samples were age matched; majority of the sample members are female. Also, alcohol consumption, tobacco exposure, and hormone receptor status were quantified, none was significantly different. Sample description has been published previously [8,9,10,21].

Subjects		Control *n* = 20 (%)	Cancer *n* = 105 (%)
Sex	Male	0	3 (2.8)
	Female	20 (100)	102 (97.2)
Age Distribution (years)	Range	52–87	27–92
Age (years)	Mean ± SD	66 ± 11	64 ± 12
Smoking Habit	Smokers	1 (5)	3 (2.8)
	Pack Year (PY)	40	56 ± 11
Alcohol Consumption	Social/Occasional Drinker	5 (25)	29 (27.62)
	Regular Drinker	0	3 (2.8)
Estrogen Receptor (ER) Status	Positive		80 (76)
	Negative		19(18)
Progesteron Receptor Status (PR) Status	Positive		66 (62.9)
	Negative		33 (31)
Human epidermal growth factor receptor 2 (HER2) Status	Positive		21 (20)
	Negative		68 (64.8)
ER, PR, HER2 (Triple) Negative	Negative		10 (9.5)

**Table 2 cancers-11-00938-t002:** Percentage of samples that show higher expression of the two miRNAs (negative delta Ct values) in various stages of tumour and hormonal receptors status. We conducted a Z score analysis by comparing proportions. The analysis shows the proportion of miR526b high expression samples were more in tumor grade II and III; however, this was not significant. For miR655, there was no difference in the distribution of high expression across tumor grades recorded. A higher proportion of samples showing high miR526b and miR655 expression can be seen in ER positive samples; however, the difference is not statistically significant, although larger sample size may increase significance. Other receptor status also shows no significant difference between presence or absence of individual receptor.

**Tumour Grade and High miRNA Expression in Cancer Samples**				
**Tumour Grade**		***n* (%)**	**miR-526b High *n* (%)**	**miR-655 High *n* (%)**
I (low-well differentiated)		7 (6.7)	0	2 (28.5)
II (intermediate-moderately differentiated)		26 (24.76)	2 (7.69)	7 (26.9)
III (high-poorly differentiated)		63 (60)	5 (7.94)	13 (20.6)
X (Unknown)		9 (8.57)	0	2 (28.6)
**Tumour Receptor Status and High miRNA Expression in Cancer Samples**				
**Receptor Status of Cancer Samples**		***n* (%)**	**miR-526b High *n* (%)**	**miR-655 High *n* (%)**
ER Status	Positive	40 (38.1)	6 (15)	10 (25)
	Negative	20 (19)	0	4 (20)
PR Status	Positive	33 (31.4)	3 (9.1)	8 (24.2)
	Negative	33 (31.4)	2 (6.1)	7 (21.2)
HER2 Status	Positive	22 (21)	2 (9.1)	8 (36.4)
	Negative	68 (64.8)	4 (5.9)	13 (19.1)
ER,PR, HER2 (Triple) Negative	Negative	10 (9.5)	0	0

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
