# Peer review of "Mir526b and Mir655 Promote Tumour Associated Angiogenesis and Lymphangiogenesis in Breast Cancer"

_cancers, 2019, doi:10.3390/cancers11070938_

Round 1

Reviewer 1 Report

The research study by Hunter et al., focuses on understanding the mechanism of miR526b and miR655 induced angiogenesis and lymphangiogenesis in breast cancer. The research paper is well written and the results presented are exciting. The efforts taken by the authors to present the research study is appreciated. 

Modifications recommended:

One of the concern in the formulated study is that the study focuses only on MCF-7 and on two cell lines generated to over express the two miRNAs investigated. While the results presented are strong and clear, the discussions need to highlight more on the role of these miRNAs in ER positive breast cancer. Also are there information on Triple negative breast cancer with respect to these miRNAs?. I recommend modifying the introduction or discussion sections to highlight ER positive breast cancer and the two miRNAs to strengthen the study. I also recommend to include discussion on Triple negative breast cancer, if the authors wishes to pursue similar study in a triple negative breast cancer cell lines. Since a cancer cell line vary vastly in their heterogeneity, it is recommended to use two or more different cell lines to understand a research question. Modifications to discussion as recommended above will strength the research article and also pave the way for future directions.  

Author Response

Reviewer 1:The research study by Hunter et al., focuses on understanding the mechanism of miR526b and miR655 induced angiogenesis and lymphangiogenesis in breast cancer. The research paper is well written and the results presented are exciting. The efforts taken by the authors to present the research study is appreciated. 

Response:  Thank you for your positive comments. We answered every concern you raised and incorporated every suggestion you gave.

Modifications recommended:

One of the concern in the formulated study is that the study focuses only on MCF-7 and on two cell lines generated to over express the two miRNAs investigated. While the results presented are strong and clear, the discussions need to highlight more on the role of these miRNAs in ER positive breast cancer. Also are there information on Triple negative breast cancer with respect to these miRNAs?. I recommend modifying the introduction or discussion sections to highlight ER positive breast cancer and the two miRNAs to strengthen the study. I also recommend to include discussion on Triple negative breast cancer, if the authors wishes to pursue similar study in a triple negative breast cancer cell lines. Since a cancer cell line vary vastly in their heterogeneity, it is recommended to use two or more different cell lines to understand a research question. Modifications to discussion as recommended above will strength the research article and also pave the way for future directions.  

Responses: Thank you for your feedback. We understand and agree with the concern you expressed regarding ER-positive breast cancer. This is an excellent suggestion. As per your suggestion, we have added a short justification as to why we chose to pick up MCF7, an ER-positive cell line, as our in vitro model to study the functions of these miRNA (pg 10 of discussion). Expression of these miRNA have been measured amongst several other cell lines with different receptor statuses (ER, PR, HER2) previously, which showed that expression of these miRNA is highest among triple negative breast cancer cell lines, and lowest among ER-positive cell line MCF7. We have previously shown that overexpression of these two miRNAs in ER-positive cell line (MCF7), results in a very aggressive phenotype and metastasis within few weeks to lung and spleen, as tested in Mouse models (Majumder et al, 2015, 2018). Thus, to investigate the gain of function, MCF7 was selected as the cell line to overexpress these miRNAs and investigate their roles in angiogenesis and lymphangiogenesis.

However, further studies using other cell line types of different receptor statuses  (triple negative and knock down these miRNAs) would help to better understand the mechanisms of these miRNA across different breast cancer subtypes.

Moreover, we looked further into the tumour sample set used in our study and investigated the relationship between miRNA expression and receptor status of the demographic and found that there was no significant difference of miRNA expression across hormone statuses. To show this, we have added an additional table below our tumour set demographic (Table 2, pg 16). Although our sample set does not show a difference of miRNA expression amongst different hormone receptor statuses, miRNA expression was high in EP positive samples. It is possible that increasing the sample size may give novel insights into the heterogeneity of miRNA expression amongst breast cancer subtypes. 

Reviewer 2 Report

The manuscript by Hunter S et al., described miR-526b and miR-655 functions on angiogenesis and lymphangiogenesis. This is a well-written and interesting study to show evidence for two miRNA that controlled angiogenesis and lymphangiogenesis. The authors have shown clear results that both miR-526b and miR-655 regulate genes related to angiogenesis and lymphangiogenesis and influence tube formation and endothelial cell branching. Also, the positive correlation between these miRNAs and angiogenesis and lymphangiogenesis factors were shown by using clinical sample data. Overall, the study was well organized, however, there are several points which the authors should revise.

Comments:

1. Title: “promotes” should be “promote”.

2. In Fig.1B, authors should perform the same western blot analysis using MCF7-miR526b samples.

3. Figure 3A is NOT described in the text.

4. Figure 4, is there any reason why the authors tested tube formation with inhibitors only in MCF7-miR655 cells?

Author Response

Reviewer 2:

Top of Form

Comments and Suggestions for Authors

The manuscript by Hunter S et al., described miR-526b and miR-655 functions on angiogenesis and lymphangiogenesis. This is a well-written and interesting study to show evidence for two miRNA that controlled angiogenesis and lymphangiogenesis. The authors have shown clear results that both miR-526b and miR-655 regulate genes related to angiogenesis and lymphangiogenesis and influence tube formation and endothelial cell branching. Also, the positive correlation between these miRNAs and angiogenesis and lymphangiogenesis factors were shown by using clinical sample data. Overall, the study was well organized, however, there are several points which the authors should revise.

 Response: Thank you for your feedback and positive comments. We answered every concern you raised and incorporated every suggestion you gave.

Comments:

1. Title: “promotes” should be “promote”.

 Response: Thank you for this suggestion, we have made this change in the title. (page 1)

2. In Fig.1B, authors should perform the same western blot analysis using MCF7-miR526b samples.

Response: This is an excellent suggestion, which helps to strengthen our results, and we have added a second western blot analysis with MCF7-miR526b. (page 3, Figure 1). We also did Densitometric measurement of western blots using ImageJ and a new figure has been added (Figure 1C).

3. Figure 3A is NOT described in the text.

Response: Thank you. We have now added a reference to Figure 3A, now changed figure 4A within the text. (page 4)

4. Figure 4, is there any reason why the authors tested tube formation with inhibitors only in MCF7-miR655 cells?

Response: This is a great question. The initial plan was to perform the tube formation inhibition with both MCF7-miR655 and MCF7-miR526b conditioned media. We plated HUVEC in two different 96 well plates for each miRNA and added conditioned media from various passages of MCF7-miR526b and MCF7-miR655 as planned. However, the HUVECs became contaminated on the plate in which the MCF7-miR526b conditioned media had been applied and unfortunately because of the limited supply of the inhibitors (specifically EP4A) used in this experiment, we were unable to complete the experiment. 

Cancers EISSN 2072-6694 Published by MDPI AG, Basel, Switzerland RSS E-Mail Table of Contents Alert
Back to Top